# Natural Mitochondria Targeting Substances and Their Effect on Cellular Antioxidant System as a Potential Benefit in Mitochondrial Medicine for Prevention and Remediation of Mitochondrial Dysfunctions

**Daniel Schniertshauer \*** 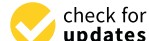**, Susanne Wespel and Jörg Bergemann**

Department of Life Sciences, Albstadt-Sigmaringen University of Applied Sciences, Anton-Günther-Str. 51, 72488 Sigmaringen, Germany

**\*** Correspondence: schniertshauer@hs-albsig.de; Tel.: +49-(0)-7571-732-8273

**Abstract:** Based on the knowledge that many diseases are caused by defects in the metabolism of the cells and, in particular, in defects of the mitochondria, mitochondrial medicine starts precisely at this point. This new form of therapy is used in numerous fields of human medicine and has become a central focus within the field of medicine in recent years. With this form of therapy, the disturbed cellular energy metabolism and an out-of-balance antioxidant system of the patient are to be influenced to a greater extent. The most important tool here is mitotropic substances, with the help of which attempts are made to compensate for existing dysfunction. In this article, both mitotropic substances and accompanying studies showing their efficacy are summarized. It appears that the action of many mitotropic substances is based on two important properties. First, on the property of acting antioxidantly, both directly as antioxidants and via activation of downstream enzymes and signaling pathways of the antioxidant system, and second, via enhanced transport of electrons and protons in the mitochondrial respiratory chain.

**Keywords:** antioxidants; dysfunctions; mitochondria; mitotropic; respiratory chain

## 1. Introduction

With an average size of only 0.75–3 μm, they are among the smallest organelles in our cells and yet account for about 36% of the volume in cardiac muscle cells [1]. The number of mitochondria in a cell varies depending on the cell type and can also be changed according to the energy requirements of a cell. For example, several thousand mitochondria work in a single heart muscle cell, whereas up to 100,000 works in a mature egg cell [2]. Relative to their weight, they produce 10,000–50,000 times more energy than the sun—energy in the form of adenosine triphosphate (ATP) [3]. In addition to energy production, mitochondria also fulfill various other functions in the cell and are involved in the regulation of the redox state, cell proliferation, heme and steroid synthesis, and apoptosis, among other things [4]. It is not surprising that when these organelles are disturbed, either by mitochondriopathies or by age-related loss of function, it has a significant negative impact on the smooth functioning of cellular processes. Therefore, in the following review, for a better understanding, we will first discuss the functioning of mitochondria and, later, possible natural mitotropic substances and their effect on the cellular system as a potential benefit in mitochondrial medicine.

### 1.1. Mitochondrial Function and Genetics

Mitochondria (Figure 1) are aerobic proteobacteria that were probably once taken up by endosymbiosis (endosymbiont theory) and are found in almost all eukaryotic cells. Endosymbiosis is thought to be the reason for the two lipid bilayers that surround mitochondria. These two membranes are referred to as the outer and inner membranes. The

outer membrane completely envelops the mitochondrion while enclosing the intermembrane space. It has a higher permeability than the inner membrane. It contains porins that are permeable to small molecules and ions up to 5 kilodaltons (kDa) in size [5]. The intermembrane space is a non-plasmatic phase with few functions. The inner membrane is adjacent to the matrix and forms cristae, the number of which varies from cell type to cell type. This leads to a significant increase in the surface area of the inner membrane, which allows for increased adenosine triphosphate (ATP) production via the electron transport chain (ETC) located here and ATP synthetases [6]. There is a brisk metabolite exchange between the cytoplasm and matrix, which is why mitochondrial membranes have a high number of transport proteins, e.g., TIM (inner membrane translocase) and TOM (outer membrane translocase). However, it is impermeable to many ions and molecules. The matrix contains ribosomes in multiple numbers, mitochondrial DNA (mtDNA) (in up to ten identical copies), and granules. It also houses all the major metabolic processes of mitochondria, including genome replication, transcription, and translation [5]. As mentioned earlier, mitochondria have a wide range of tasks and, in addition to generating ATP, perform a central role in programmed cell death (apoptosis) through the release of cytochrome c caused by a change in mitochondrial membrane permeability. In cell-fee extracts, caspase initiation was shown to be possible only in the presence of cytochrome c and dATP (deoxy-adenosine triphosphate) [7]. They also induce cell death by interrupting energy metabolism via the electron transport chain [8]. Synthesis of iron-sulfur clusters (Fe-S clusters) is considered an essential function of mitochondria. These Fe-S clusters are required as cofactors for a variety of enzyme reactions and, therefore, are of great importance for the survival of a cell [9].

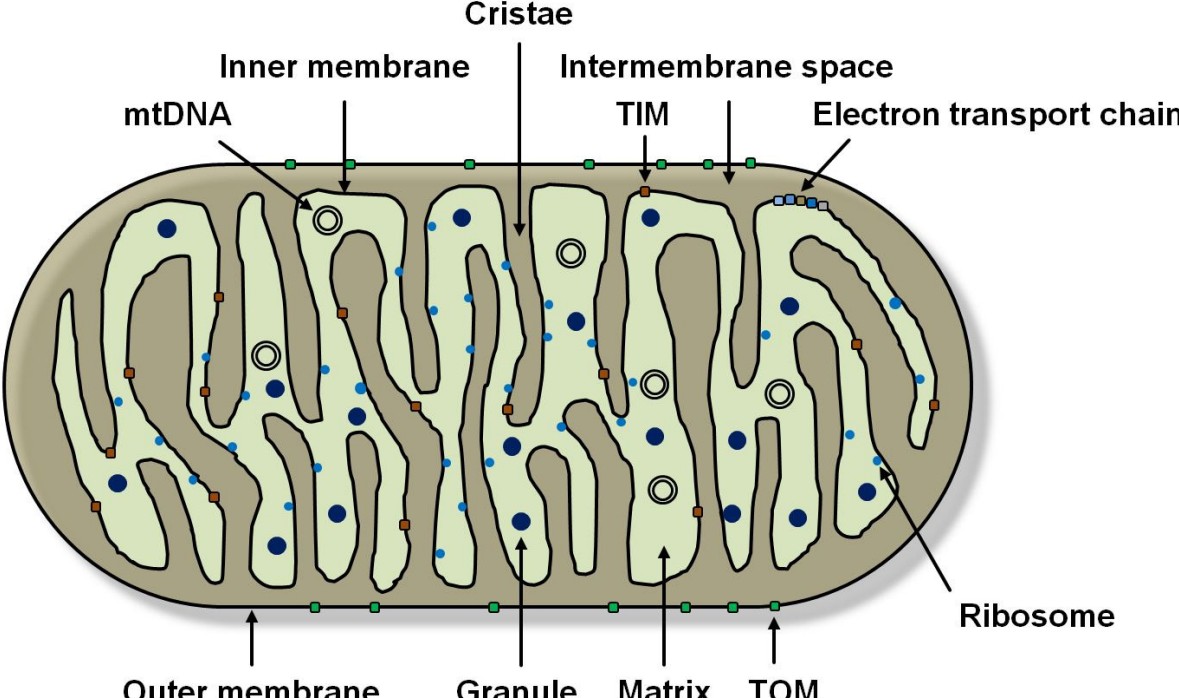

**Figure 1.** Schematic structure of a mitochondrion. Mitochondria are surrounded by two membranes defining the intermembrane space and the matrix. The inner membrane is highly folded to increase surface area and forms cristae. Complexes of the electron transport chain (ETC) are located here. Both mitochondrial membranes have a high number of transport proteins, e.g., TIM (inner membrane translocase) and TOM (outer membrane translocase). Among other things, the matrix contains mitochondrial DNA (mtDNA), ribosomes, and granules.

The 16.5 kilobases (kb) mitochondrial genome is double-stranded, circular, and has no histones or introns. In 37 genes, it encodes 2 organelle-specific ribosomal ribonucleic

acids (rRNA) related in size and sequence to bacterial rRNA, 22 transfer RNAs (tRNA), and 13 essential mitochondrial proteins [2,10]. Among them, some divergent codons were found in mitochondria. In humans, the mitochondrial code for AUA replaces the normally encoded isoleucine with methionine; AGA and AGG mean "stop" instead of coding for arginine, as normally is the case. These differences between the nuclear and mitochondrial genomes have supported the view that mitochondria in eukaryotic cells were originally symbionts of prokaryotic origin before evolving into obligate components of eukaryotic cells. The "universality of the genetic code" is one of the central laws of molecular biology. It states that the information content of the nucleotide sequence is read and transcribed in the same way by all living organisms. However, mitochondria are also a notable exception to this central point. Mitochondria are generally semi-autonomous organelles and, therefore, have a genome that encodes only a few mitochondrial proteins. Thus, most regulatory factors and structural proteins are encoded by nuclear DNA and must be transported from the cytoplasm to the mitochondria after their synthesis. Even parts of the mitochondrial membrane and many of the enzymes required in them are encoded in the nucleus and must therefore be imported into these organelles. Therefore, it has long been assumed that mitochondria are not viable without the nucleus. However, it is now known that functional mitochondria are not only found in the non-nucleated platelets but can also occur in high numbers as cell-free mitochondria in the blood [11]. During cell division, mitochondria distribute to daughter cells and proliferate by division, which occurs independently and autonomously of the cell. The mitochondria are maternally inherited, which prevents recombination of the mtDNA, e.g., in meiosis [2].

### 1.2. The Respiratory Chain

The generation of ATP by oxidative phosphorylation, which occurs at the inner mitochondrial membrane, is achieved by the transport of electrons along complexes I to IV and the mobile components ubiquinone (CoQ) and cytochrome c of the ETC (Figure 2). The latter two serve as electron and proton transfer agents between these complexes. The largest enzyme complex of the respiratory chain, complex I, also known as NADH-ubiquinone oxidoreductase, catalyzes the transport of two electrons from the 1,4-dihydronicotinamide adenine dinucleotide (NADH) to CoQ and the transport of four protons ($H^+$) from the matrix to the intermembrane space (Q cycle). This contributes to the formation of a proton gradient across the inner membrane. Reduced ubiquinol ($CoQH_2$), a two-electron carrier that can diffuse freely in the lipid bilayer of the inner membrane, is formed [5].

At complex II (succinate dehydrogenase), also part of the citrate cycle, the oxidation of succinate to fumarate takes place, leading to electron transfer to CoQ, resulting in $CoQH_2$. In addition to three iron-sulfur clusters (Fe-S clusters), complex II has a flavin adenine dinucleotide (FAD) prosthetic group and a succinate binding site. Electrons travel from the succinate to CoQ via FAD and the Fe-S clusters [5]. Subsequently, at dimeric complex III, the ubiquinone cytochrome c oxidoreductase, electron transfer from $CoQH_2$ to cytochrome c occurs with the translocation of four protons from the matrix to the intermembrane space. Cytochrome c, a one-electron carrier, further transports these electrons from complex III to complex IV and donates them to the two divalent copper ions ($Cu^{2+}$) of cytochrome c oxidase (CuA) located in subunit II. This subunit has two heme groups (heme a) as a prosthetic group [5]. Complex IV, cytochrome c oxidase, catalyzes electron transport from cytochrome c to oxygen ($O_2$) and reduces it to two molecules of water ($H_2O$). Subunit I of complex IV has two heme groups (heme a and heme a3) and another $Cu^{2+}$ ion (CuB). Heme a3 and CuB form another binuclear center. Thus, electron transport from cytochrome c to $O_2$ occurs via the CuA center, the heme a, and the heme a3-CuB center. The energy released during the reduction in $O_2$ to $H_2O$ is used to pump an additional four $H^+$ per oxygen molecule from the matrix via complex IV and the inner membrane into the intermembrane space. To prevent the formation of reactive oxygen species (ROS) (see infobox) in the process, the heme-a3 CuB center must be first reduced with two electrons. Only under this condition is it possible for the $O_2$ to bind. In this way, the oxygen can be reduced

directly to peroxide before it can be broken down into its individual atoms. The proton gradient established during this electron transport catalyzes ATP synthesis from adenosine diphosphate (ADP) and inorganic phosphate ($P_i$) by a proton flux from the intermembrane space back into the mitochondrial matrix by means of $F_0F_1$-ATP synthase (complex V). In this process, a high electrical voltage must be maintained across the mitochondrial membrane at field strengths of approximately 1,000,000 volts/cm. The inner mitochondrial membrane acts as a kind of insulator, whereas in air, an electric spark would already jump at 10,000 volts/cm and, thus, lead to a short circuit [12]. ATP synthase in mitochondria is an F-type ATP synthase and consists of two components, $F_1$ and $F_0$. $F_1$ is a peripheral membrane protein with nucleotide binding sites, whereas $F_0$ is integrated into the membrane and has a proton channel [5]. The $F_0$ subunit consists of three major subunits, a, b, and c, and six other subunits. $F_1$ is a hexamer consisting of three α- and three β-subunits. $F_1$ and $F_0$ are connected by a peripheral and a central stalk consisting of the γ-, δ-, and ε-subunits. The γ-subunit brings the c-subunits of $F_0$ into contact with the δ- and ε-units. According to the chemiosmotic model, the proton motive force drives the ATP synthase. The returning protons cause rotational motion of the c-ring of $F_0$ and the γ-, δ-, and ε-subunits of the central stalk. The resulting rotational motion is transferred to the $F_1$ moiety. This causes ATP formation in the $F_1$ part via a conformational change. The peripheral stalk thereby prevents the entire $F_1$ subunit from rotating. When the γ-subunit rotates completely, three ATPs are formed [5]. The transfer of electrons and the ATP synthase are coupled and do not function without the other part [13]. In addition to ATP synthase, $H^+$ can also be pumped back into the matrix via other mitochondrial membrane proteins called uncoupling proteins (UCPs). This leads to the uncoupling of the electron transport chain from ATP synthase [14].

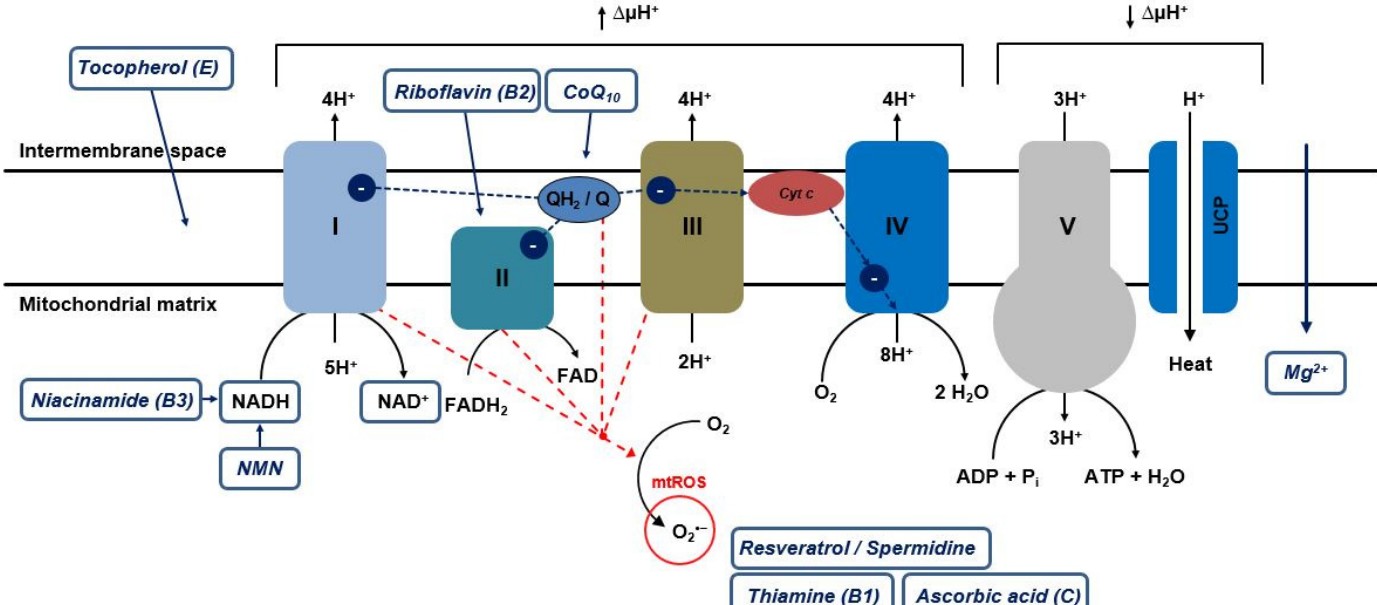

**Figure 2.** Simplified representation of the mitochondrial respiratory chain and targets of selected mitotropic substances. The inner mitochondrial membrane and the compartments of the respiratory chain are shown. Complexes I–IV generate a proton gradient ($\uparrow\Delta\mu H^+$) so that ATP can be synthesized via complex V (ATP synthase) ($\downarrow\Delta\mu H^+$). Electrons move from complex I via complex II and ubiquinol to complex III. Cytochrome c transports the electrons from complex III to complex IV. UCP (uncoupling protein) allows protons to pass through the mitochondrial membrane without chemiosmotic coupling, thus uncoupling the respiratory chain from ATP generation and serving, among other things, thermogenesis.

### 1.3. Reactive Oxygen Species (ROS)

Based on 80% of the oxygen consumed by mammals, which is utilized in the mitochondria, 1–2% is converted into reactive oxygen species, so-called ROS [15]. ROS refers to

compounds of molecular oxygen that are a by-product of cellular metabolism [16]. They exist as radicals, ions or molecules that possess an unpaired electron in their outer electron shell [17,18]. Due to these properties, these compounds have high reactivity and chemical aggressiveness and, therefore, are capable of oxidizing and, thus, damaging cellular biopolymers—such as DNA, proteins, and lipids [17–19]. The group of ROS includes radical ROS, such as the highly reactive hydroxyl radical ($^\bullet$OH) or the superoxide anion ($O_2^{\bullet-}$), which is considered the precursor of almost all ROS [20]. Non-radical ROS are represented by hydrogen peroxide ($H_2O_2$), hydroperoxide (ROOH), or singlet oxygen ($^1O_2$). ROS are found in all aerobic organisms and, due to a nonlinear dose–response relationship, fulfill important physiological functions at low concentrations, such as signal transduction in the brain. At high pathological concentrations, ROS contributes to the development of various diseases, such as oxidative stress [16]. Mitochondria are therefore considered to be the main source of these compounds, especially mitochondrial superoxide (mtSO), a chemical compound containing the oxygen-derived superoxide anion. As ROS and mtSO are mainly unwanted byproducts of the respiratory chain (Figure 2), the formation of these aggressive compounds is correlated with the synthesis of adenosine triphosphate (ATP) only in the last step in the respiratory chain. The formation of these compounds depends on the respiratory state, i.e., when the mitochondrial membrane potential is high, and the ADP level is low, the production of ROS is high. When the membrane potential is low, and the ADP level is high, the production of ROS is low [15]. This is caused by an electron leak between complexes I and III [18,21,22]. Due to the relatively high ROS production in mitochondria and the fact that mtDNA is more susceptible to external influences (such as ROS) due to the lack of protective histone proteins, the genetic information of these organelles is oxidatively damaged about ten times more frequently than nuclear DNA. Here, the mutation rate in the absence of external influences is about one nucleotide exchange per $1.0 \times 10^9$ base pairs in each round of replication [15,23]. Furthermore, oxidative mtDNA damage in the respiratory chain leads to an increase in this electron leak and, thus, in the number of radicals, which in turn contribute to a further increase in the mutation rate and, thus, to mtDNA damage [24–27]. According to the "mitochondrial theory of aging", these mitochondrial ROS (mtROS) perform a very important role in aging processes and age-associated diseases together with other oxidative mtDNA damage and a limited repair capacity of mitochondria [28]. However, the details, such as the exact amount of mtROS, remain controversial [29,30], although it was already shown, in the 1980s and 1990s, that mtDNA is severely damaged by ROS [31,32].

### 1.4. Mitochondria in the Process of Aging

The aging process is characterized by an increase in age-related disorders and severe diseases. Due to their role in oxidative phosphorylation and, thus, in the production of ATP, which is crucial for many cellular processes, one reason for these disorders and diseases could be found in defective mitochondria [30,31]. Although aging is a multifactorial and thus much more complex process than previously thought, only the mitochondrial mechanisms that play a role will be discussed here. In the "free radical theory of aging" (FRTA), developed by Harman in 1956, he identified free radicals as the cause of aging processes because they damage molecules that are important for many important cellular functions, such as DNA, RNA, and also proteins and lipids. After the discovery of the mitochondrial genome, he expanded his theory to the "mitochondrial free radical theory of aging" (MFRTA), which deals with the age-related loss of mitochondrial function due to an increasing accumulation of oxidative damage. This theory is also known as the "mitochondrial theory of aging" [32,33]. According to this theory, the accumulation of ROS that damage mitochondrial DNA and proteins can lead to mitochondrial dysfunctions within the electron transport chain. Due to these dysfunctions, more ROS are formed, which in turn damage mtDNA. The accumulation of mtDNA damage then leads to new defects in the electron transport chain, increased ROS production, and higher oxidative stress. This eventually leads to a vicious cycle of ROS production [24–27]. In other theories dealing with

the effects of early mitochondrial dysfunction, energy metabolism performs a central role. For example, Prinzinger's "maximum slope theory" deals with the correlation of lifespan with cell energy production, and Prinzinger states that aging is a consequence of the limited ability of mitochondria to produce energy [34]. In another theory, the defective power plant model, Krutmann and his colleagues discuss the role of mtDNA mutations and disorders in the respiratory chain as factors for dysfunction in mitochondria and, thus, for limited lifespan [35]. Ultimately, what all theories have in common is that aging is considered, among other things, to be a consequence of dysfunction in mitochondrial function and that this organelle thus performs a very important role in aging processes and age-associated diseases [36–40]. This was recently confirmed in experiments performed on biopsies of the human epidermis. Here, for the first time, a decrease in mitochondrial respiration of about 10% per decade and a decrease in ATP production with increasing donor age could be observed directly ex vivo in human tissue, consistent with the "mitochondrial theory of aging" [41].

## 2. Mitochondrial Dysfunctions—Clinical Relevance

Mitochondrial diseases, also known as mitochondriopathies, are diseases in which there is a defect in these organelles. To date, about 50 of these diseases are known [30]. The reason for this is often found in the cell nucleus and in the mitochondrial genome. Due to its structure (lack of histones) and its localization in the direct vicinity of the respiratory chain, mtDNA is particularly susceptible to damaging influences, such as ROS. As a result, the mutation rate in the mitochondrial genome is ten times higher than in nuclear DNA. However, due to a large number of mtDNA copies per cell, mitochondria—in contrast to the nucleus—have the ability to eliminate even larger portions of defective DNA and replace them with intact copies [23,42]. However, if damage occurs within the cell in approximately two-thirds of all mitochondria, resulting in mitochondrial failure, cellular energy requirements are no longer adequately secured and severe mitochondrial dysfunction and associated diseases are to be expected. These include diabetes, cardiovascular disease, cancer, and age-related neurodegenerative diseases, such as Alzheimer's disease and Parkinson's disease, that predominantly involve cells with high energy demands and consequently increased numbers of mitochondria [34,35,42]. Initial indications of the loss of intact mitochondria may include memory and concentration difficulties, declining physical fitness, and decreasing vision, smell, and hearing. The thousands of copies of mtDNA in a cell are nearly identical at birth. In contrast, patients with mitochondrial dysfunction have a mixture of mutant and wild-type mtDNA in each cell [43,44]. Furthermore, the extent of mutant DNA varies within organs and tissues of the same individual [45]. This is one explanation for the high diversity of phenotypes in patients with mitochondrial dysfunction [46]. In the development of mitochondriopathies, the focus has so far been on the inheritance of genetic factors, but for some time now, other harmful influences, such as the effects of environmental factors, e.g., UV radiation [28], but also drugs, have also come into focus as triggers of mitochondrial dysfunctions. Since, according to the endosymbiont theory, mitochondria are descended from bacteria that have been taken up by eukaryotic cells, these drugs are mainly antibiotics, but also chemotherapeutic agents and methylphenidate, which, for example, blocks complex I of the mitochondrial respiratory chain and, thus, increases mtSO production. In the long term, however, inflammation can also lead to damage in the mitochondria since this is always accompanied by the release of ROS. Likewise, psychological suffering in the body leads to the formation of ROS and, thus, to neurostress, which can also cause mitochondrial dysfunction [47]. The presence of a high concentration of ROS or an increase in their concentration leads to a shift in the intracellular balance between oxidation and reduction in favor of oxidation processes. This favors the formation of more ROS and prevents quantitatively sufficient inactivation of these compounds by the antioxidants available to the cells, further shifting the balance [19]. The disturbance of the intracellular balance between the formation of ROS and its inactivation is called oxidative stress [19,48]. To restore this balance, the use of

mitotropic substances—substances that positively influence mitochondrial function—has been proposed for some time.

## 3. Mitotropic Substances

Mitotropic substances, or so-called mitoceuticals, describe substances whose mode of action is directed at the bioenergetic processes of the mitochondria and are, in part, essential for their function. The term mitoceuticals, created by Dr. Franz Enzmann, encompasses several groups of mitotropic substances, which have a versatile potential and form the basis for a preventive and therapeutic approach in mitochondrial medicine. They act non-specifically and reach all mitochondria in the organism. The best-known substances include electron and proton carriers, such as ubiquinol/ubiquinone (coenzyme $Q_{10}$; $CoQ_{10}$), vitamins, minerals, and micronutrients. Mitoceuticals are substances that are part of the natural metabolic system of a cell anyway. They are used both preventively and therapeutically for accompanying treatment. In both cases, the aim is to compensate for an existing mitochondrial dysfunction. In the meantime, there are a large number of mitotropic substances and even more studies on them. In order to provide an overview of the various substance classes and their targets, the most important natural substances and structures are described below. For better limitation, only natural substances were selected, which also act in a non-directional manner. Anticancer agents, such as mitocans, an acronym made up of the terms mitochondria and cancer, were not included because they are, in part, selective for malignant tissues [49]. The classification into the individual groups is completed according to typical linguistic usage but is in part across groups.

### 3.1. Electron and Proton Carrier

Electron and proton carriers are proteins and enzymes with the ability to transport electrons or protons and, in some cases, both. This is often due to their redox-active character [42]. The UCPs mentioned in point 1.2 The respiratory chain, which separates oxidative phosphorylation from ATP synthesis, also belong to the group of (mitochondrial) proton carriers [14]. Electron and proton carriers are essential for the generation of ATP by oxidative phosphorylation since they ensure the necessary transport of electrons along complexes I to IV and the mobile components ubiquinone (CoQ) and cytochrome c and the establishment of a proton gradient across the inner mitochondrial membrane [42]. In addition to the mentioned substances, such as $CoQ_{10}$ and UCPs, nicotinamide mononucleotide (NMN), a molecule from the family of B3 vitamins, for example, also belongs to this group.

### 3.1.1. Coenzyme $Q_{10}$

$CoQ_{10}$ is a ubiquitous endogenous quinone derivative found in the biological membranes of the majority of all cells in the body. It is also present as an antioxidative component in circulating lipoproteins [15,42]. Structurally, $CoQ_{10}$ consists of a redox-active 2,3-dimethoxy-5-methylbenzoquinone ring and an isoprenoid side chain coupled to it at position 6. Depending on the sequence of functional groups attached to the quinone ring, physiologically irrelevant conformations (cis-conformation) can be distinguished from biologically active isomers (trans-conformation) [50]. The number of linked dihydroisoprene units differs in different organisms and varies from six to ten dihydroisoprene units [42]. In humans, the chain is predominantly made up of ten monomers, which is why the lipid is accordingly referred to as $CoQ_{10}$. 7–9% of the coenzyme is present in the human organism in the form of $CoQ_9$ [15]. The 1,4-benzoquinone ring of ubiquinone and the 1,4-benzohydroquinone of ubiquinol represents the active functional group of the lipid. Via this redox-active structural element, electrons are taken up and released as a result of reduction and oxidation processes. In addition to its antioxidative properties, the lipophilic isoprenoid side chain serves to anchor the coenzyme in biological membranes, thereby increasing the fluidity and permeability of the membrane [15].

*Molecular mechanism/Biological function—Pleiotropic effects of coenzyme Q$_{10}$*

In view of these versatile properties of CoQ$_{10}$, it is not surprising that this results in quite a few effects at the cellular level and that this coenzyme thus fulfills numerous functions in the human organism. For this, the term Q$_{10}$ pleiotropy is proposed for the first time. Due to its redox-active character, the ability to transport protons and electrons by redox reactions, CoQ$_{10}$ performs a central role in the course of the mitochondrial respiratory chain. The lipid located in the inner mitochondrial membrane acts as a transport system, which transports electrons from complexes I and II through the hydrophobic membrane interior to complex III of the respiratory chain [42]. CoQ is reduced to ubiquinol CoQH$_2$ by the uptake of two electrons and two protons. In the opposite direction, CoQH$_2$ can be oxidized to CoQ by the release of two electrons and protons (Figure 2). This redox reaction pair, consisting of 2,3-dimethoxy-5-methyl-1,4-benzoquinone and 2,3-dimethoxy-5-methyl-1,4-benzohydroquinone, represents the Q cycle and reflects the role of Q$_{10}$ as an electron carrier. Extramitochondrial electron transport (e.g., in lysosomes) also occurs via this coenzyme [15]. First, the electrons originating from the cellular redox equivalents NADH and the hydroquinone form of the flavin adenine dinucleotides (FADH2) are transferred to CoQ by complex I or complex II. CoQH$_2$ then transfers the electrons to cytochrome c via the enzyme ubiquinone-cytochrome c oxidoreductase (complex III). Thus, the electron flow is used to pump protons from the mitochondrial matrix into the intermembrane space to create an electrochemical proton-gradient across the inner mitochondrial membrane. This gradient is used to generate ATP using complex V (Figure 2) [51]. The diffusion of CoQ$_{10}$ is significantly faster than its conversion at the respiratory complexes, which is why CoQ$_{10}$ behaves like a mobile diffusing CoQ$_{10}$ pool within the inner mitochondrial membrane [52]. In addition to the naturally occurring antioxidants carotenoids, estrogens, and tocopherols, CoQ$_{10}$ performs a special role in the human body's antioxidant system because it is the only fat-soluble antioxidant that can be synthesized endogenously [15,42,53]. As an antioxidant, the reduced form CoQH$_2$ effectively prevents oxidation of DNA, lipids, and proteins [54]. The extraordinary efficiency as an antioxidant is due to the numerous cellular mechanisms for the regeneration of CoQH$_2$ and the localization of CoQ$_{10}$ [42]. Since cell-damaging ROS are mainly formed in the inner membranes of the mitochondria (site of the respiratory chain), the CoQH$_2$ localized there can capture the oxidants formed directly at the site of their formation, neutralize them as a result of reduction processes and thus protect cellular structures and proteins and DNA from oxidative damage [15,42]. Furthermore, its antioxidant property is associated with the ability to interfere with the initiation and proliferation of lipid peroxidation (a radical oxidation of unsaturated fatty acids) [42]. CoQ$_{10}$, which is oxidized to CoQ, is continuously converted back to its reduced form, CoQH$_2$, by enzymatic processes in order to maintain its antioxidative properties [42]. This ensures that the CoQ$_{10}$ contained in blood plasma usually consists of about 95% CoQH$_2$ and about 5% CoQ [55].

*Studies on the efficacy of coenzyme Q$_{10}$*

Most studies to date have been completed on CoQ$_{10}$. For example, studies from 2007 already demonstrated a neuroprotective effect of CoQ$_{10}$ against the loss of dopamine induced by 1-methyl-4-phenyl-1,2,3,6-tetrahydropyridine (MPTP)-in patients with Parkinson disease (PD), both with CoQ$_{10}$ and with reduced CoQ$_{10}$ [56]. MPTP, as a precursor of the mitochondrial respiratory chain inhibitor MPP+ (1-methyl-4-phenylpyridinium), leads to the destruction of dopaminergic cells in the human brain (in the substantia nigra). Neuroprotective effects against dopamine loss, loss of tyrosine hydroxylase neurons, and induction of alpha-synuclein inclusions in substantia nigra pars compacta were also demonstrated. The finding that CoQ$_{10}$ is effective in a chronic dosing model of MPTP toxicity is of particular interest, as this may be of greater relevance to PD. It is further evidence that CoQ$_{10}$ administration is a promising therapeutic strategy for the treatment of PD as well [56].

Patients with fibromyalgia, a chronic pain disorder whose exact causes are still unknown, also show positive effects when treated with $CoQ_{10}$. Eleven fibromyalgia patients were randomly assigned to two treatment groups after a two-week wash-out period. One group received Pregabalin (antiepileptic drug used to treat epilepsy, neuralgia, and generalized anxiety disorder) with $CoQ_{10}$, while the second group received only Pregabalin with placebos for 40 days each. Subsequently, patients in the $CoQ_{10}$ group were switched to placebo, and patients in the placebo group were switched to $CoQ_{10}$ for an additional 40 days [57]. Using isolated peripheral blood mononuclear cells (PBMCs) to examine mitochondrial oxidative stress and inflammation levels, it was confirmed that $CoQ_{10}$ supplementation was effective in reducing greater pain, anxiety and brain activity, mitochondrial oxidative stress and inflammation. $CoQ_{10}$ also increased patients' reduced glutathione levels and superoxide dismutase (SOD) levels. These results suggest that $CoQ_{10}$ supplementation provides further benefit in alleviating pain sensation in fibromyalgia patients compared to those treated with Pregabalin, possibly by reducing mitochondrial dysfunction [57].

Mitochondrial dysfunction is also a fundamental abnormality in the vascular endothelium and smooth muscle of patients with pulmonary arterial hypertension (PAH) [58]. As coenzyme Q (CoQ) is essential for mitochondrial function and efficient oxygen utilization as an electron transporter in the inner mitochondrial membrane, it was hypothesized that CoQ would improve mitochondrial function and benefit PAH patients. To test this, oxidized and reduced CoQ levels, cardiac function by echocardiogram, mitochondrial functions of heme synthesis, and cellular metabolism were examined in PAH patients compared with healthy controls at baseline and after 12 weeks of oral CoQ supplementation. CoQ levels were similar in PAH patients and control subjects and increased in all subjects receiving CoQ supplementation. PAH patients had higher CoQ levels than control subjects with supplementation and a tendency toward a higher ratio of reduced to oxidized CoQ. Cardiac parameters improved with CoQ supplementation [58]. Hemoglobin increased, and red cell distribution width (RDW) decreased in PAH patients with CoQ, whereas hemoglobin decreased slightly and RDW did not change in healthy control subjects. In conclusion, CoQ improved hemoglobin and erythrocyte maturation in PAH patients [58]; again, this is most likely attributable to an improvement in mitochondrial function.

It is hypothesized that the administration of supplemental $CoQ_{10}$ leads to enhanced electron transport and, thus, an increase in respiratory parameters. A correlation between $CoQ_{10}$ concentration and respiration rate has already been reported in the literature, leading to the assumption that physiological concentrations of $CoQ_{10}$ alone cannot saturate the respiratory chain [59,60]. Therefore, it was expected that administration of supplemental $CoQ_{10}$ could increase mitochondrial respiration. Studies have shown that $CoQ_{10}$ is able to maintain mitochondrial membrane potential in the context of short-term induced damage after UVA irradiation, reducing the degree of mitochondrial dysfunction and, thus, leading to faster regeneration of energy metabolism in human fibroblasts [61]. However, an improvement in mitochondrial parameters was not only shown at the cellular level after UV irradiation but also in measurements of the same parameters directly in biopsies of human epithelial tissue. Here, a decrease in mitochondrial respiration of about 10% per decade and a decrease in ATP-linked respiration with increasing donor age were observed, consistent with the "mitochondrial theory of aging." However, when the reduced form of $CoQ_{10}$, ubiquinol, was added to the biopsies, regeneration of mitochondrial respiration and almost all major respiratory chain biomarkers occurred [41]. The fact that there is a significant effect of administered $CoQ_{10}$ on respiratory parameters suggests that it is likely caused by an increase in the electron transport chain. Here, it is suspected that exogenously supplied $CoQ_{10}$ counteracts a possible bottleneck (electron clamp) of electron migration in complex I/II and III; thus, increasing transport between the complexes [41].

As mentioned above, according to the mitochondrial theory of aging, there is an accumulation of ROS, which damage mitochondrial DNA and proteins, leading to mitochondrial dysfunction within the electron transport chain. Due to these dysfunctions, more

ROS are formed, which in turn damage mtDNA. One of the most common damages to mtDNA is the oxidative damage 7,8-dihydro-8-oxoguanine. Accumulation of this and other DNA damage can lead to dysfunction in the electron transport chain and, ultimately, to mitochondrial dysfunction. Since, in addition to the formation of ROS, there are also numerous exogenous effects, such as UV radiation, which induce the formation of precisely this damage, a quantitatively sufficient repair of all occurring oxidatively damaged guanine bases is, therefore, often only partially possible, especially in advanced age. Here, however, it was shown that the structural properties of $CoQ_{10}$ lead to an increase in the activity of the enzyme responsible for the repair of this damage—human 8-oxoguanine DNA glycosylase 1 (hOGG1) [62]. This results in a change in the bifunctionality of this enzyme and a direct interaction between $CoQ_{10}$ and 8-oxoguanine DNA glycosylase 1 [62]. Interactions between hOGG1 and other proteins have already been demonstrated. For example, hOGG1 interacts with the mitochondrial protein NDUFB10 (NADH dehydrogenase [ubiquinone] 1 beta subcomplex subunit 10) in complex I of the respiratory chain [63]. In addition, supplementation with $CoQ_{10}$ is thought to promote the activity of other enzymes that contribute to the neutralization of ROS, including SOD and glutamate dehydrogenase (GDH). These enzymes, which are primarily present in mitochondria, require antioxidants as cofactors to neutralize ROS [64]. An increase in these enzymes has been demonstrated in rat cancer cells during tamoxifen therapy with additional $CoQ_{10}$ administration [65]. Another possibility for $CoQ_{10}$-induced effects on ROS formation after oxidative stress could be so-called uncoupling proteins (UCPs). $CoQ_{10}$ is involved in preventing excessive mtROS production as a cofactor for mitochondrial UCPs [66,67]. In particular, UCP2 and UCP3 are responsible for maintaining low ROS levels. When these proteins are activated, this leads to an uncoupling of oxidative phosphorylation and, thus, to a reduction in the proton gradient across the inner mitochondrial membrane. This reduces ROS formation and, thus, the probability of interaction of electrons with oxygen [14]. UCPs are stimulated by activators that are normally composed of ROS themselves [15]. This negative feedback mechanism represents a regulatory mechanism and counteracts overproduction of ROS [14].

### 3.1.2. NMN/$NAD^+$/NADH

Nicotinamide mononucleotide (NMN) is a molecule that can be found in all life forms. It is the immediate precursor of nicotinamide adenine dinucleotide (NAD) and, therefore, is involved in the biosynthesis of $NAD^+$ and its reduced form NADH in all living organisms. NAD is a hydride ion (two-electron/one-proton) transferring coenzyme involved in numerous redox reactions of cellular metabolism [68]. NMN is structurally composed of a nicotinamide group, a ribose and a phosphate group [47]. At the molecular level, it is a ribonucleotide, which is a basic structural unit of the nucleic acid RNA. NMN exists in α- and β-anomeric forms, with the β-form being the active anomer. NMN is formed naturally from B vitamins by a reaction catalyzed by the enzyme nicotinamide phosphoribosyltransferase (NAMPT). NAMPT binds nicotinamide (a B3 vitamin) to a sugar phosphate called PRPP (5′-phosphoribosyl-1-pyrophosphate). NMN can also be prepared from "nicotinamide riboside" (NR) by adding a phosphate group [69].

*Molecular mechanism/Biological function*

NMN is mainly found in the nucleus, mitochondria, and cytoplasm, where it is first synthesized to 1,4-dihydronicotinamide adenine dinucleotide (NADH), then ultimately converted to $NAD^+$ in complex I of the respiratory chain (Figure 2). NMN is therefore considered a key component in increasing cellular $NAD^+$ levels [68]. Although NMN was initially considered only as a cellular energy source and intermediate in $NAD^+$ biosynthesis, attention is now focused on its special role as a precursor of $NAD^+$ since this metabolically important redox coenzyme performs a crucial role in a variety of biological processes in the body [70]. $NAD^+$ acts as a coenzyme in metabolic processes, such as glycolysis, the TCA (tricarboxylic acid cycle) cycle and the electron transport chain. $NAD^+$ is also required by cells for the regulation of gene expression and DNA repair. It thereby increases cell viability by reducing cell damage and reducing apoptosis [68,70]. $NAD^+$ is also a cofactor

for processes in mitochondria, for sirtuins, and for the poly(ADP-ribose) polymerases PARP (among others, an enzyme for the repair of single-stranded DNA breaks) [71]. Sirtuins are $NAD^+$-dependent deacylases that perform a key role in the response to nutritional and environmental perturbations, such as fasting, DNA damage, and oxidative stress [72]. Each single repair of a DNA single-strand break consumes $NAD^+$ molecules. Cells experiencing a high number of DNA single-strand breaks are often deficient in this substance [73]. Thus, increasing $NAD^+$ synthesis through increasing the availability of $NAD^+$ precursors, such as NMN, can help maintain cellular integrity and function under these conditions. Since not only $CoQ_{10}$ levels, but also $NAD^+$ levels decrease with age, it seems understandable to counteract this with an additional administration of $NAD^+$. A drop in $NAD^+$ levels during aging could be a vulnerability that also causes defects in nuclear and mitochondrial functions and leads to many age-related diseases [72].

*Studies on the efficacy of NMN/$NAD^+$/NADH*

Since, with advancing age, not only the $CoQ_{10}$ levels but also the $NAD^+$ levels decrease, it seems understandable to counteract this with an additional administration of $NAD^+$. A decrease in the $NAD^+$ level during the aging process could be a weak point, which also causes defects in the nuclear and mitochondrial functions and leads to many age-related diseases. Restoring decreased $NAD^+$ levels can significantly improve age-related dysfunction and counteract many age-related diseases, including neurodegenerative diseases. Thus, it has been shown that a combination of sirtuin activation and $NAD^+$ administration can be an effective anti-aging intervention [72]. A systemic decrease in $NAD^+$ is considered a likely explanation for why aging effects on sirtuins. Importantly, intermediate $NAD^+$ supplementation appears to restore $NAD^+$ levels in both the nucleus and mitochondria of cells. In general, activation of sirtuins triggers nuclear transcriptional programs that upregulate mitochondrial metabolism and associated resistance to oxidative stress [72].

In one study, aging was shown to trigger SIRT1 inactivation, which was reversed by NMN [72]. Additionally, based on the SIRT1/Nuclear factor erythroid 2-related factor2(Nrf2)/Heme oxygenase-1 (HO-1) signaling pathway, NMN was shown to increase viability in human corneal epithelial cell treated with high glucose by reversing cell damage, reducing apoptosis, and increasing cell migration [68].

In another study, mitochondrial deficiency in complex I of the ETC resulted in depletion of mitochondrial $NAD^+$ due to accumulation of NADH, inactivation of mitochondrial SIRT3, and severe cardiac damage [72].

Treatment of heart failure (HF) patients with nicotinamide riboside (NR), an NAD precursor, for 5 to 9 days demonstrated that HF-related decreased respiratory capacity and increased expression of pro-inflammatory cytokines can be attenuated by oral administration of NR [74]. There was an improvement in mitochondrial respiratory parameters in PBMCs and a reduction in gene expression of pro-inflammatory cytokines in four subjects with HF [74]. This suggests that systemic inflammation in patients with HF is causally related to mitochondrial function in PBMCs. Increasing $NAD^+$ levels may have the potential to improve mitochondrial respiration and attenuate pro-inflammatory activation of PBMCs in HF [74].

### 3.2. Vitamins

Vitamins are organic compounds that an organism needs not as energy carriers but for other vital functions. They are involved in many reactions, influence the immune system, for example, and are indispensable in building cells of all human tissue. With the exception of vitamins D and B3, which the human body synthesizes itself, all other vitamins must be taken up completely with food, as they are essential substances. Some vitamins are supplied to the body as precursors, so-called provitamins, which are first converted into their active form in the body [75]. This review focuses on the class of B, C, and E vitamins because of their importance for mitochondrial medicine. Chemically, vitamins do not form a uniform group of substances. Since they are quite complex organic molecules, they do not occur in inanimate nature. They must first be formed by plants,

bacteria, or animals [76]. According to their chemical function, the 13 known vitamins can be classified as free radical scavengers, coenzymes, or precursors of messenger substances and dyes. A further subdivision is made into fat-soluble (lipophilic) and water-soluble (hydrophilic) vitamins. In contrast to other biomolecules, such as proteins or nucleic acids, vitamins do not have a uniform structure and differ greatly from one another structurally. Thus, among the vitamins, there are steroids, isoprenoids, pyrimidines, pyridines, sugar acids, and urea derivatives. For better differentiation, several related compounds with comparable properties are grouped under a single vitamin [77].

*Molecular mechanism/Biological function*

Vitamins are needed by the body to maintain its many vital functions. Depending on the vitamin, the daily requirement is between 20 μg (vitamin D) and 100 mg (vitamin C) and is influenced by individual circumstances, such as body weight, the amount of physical work or even existing diseases. If the requirement is not met or exceeded, diseases may occur, which are called vitaminoses. The essential functions of vitamins include controlling the metabolism of protein, carbohydrates, and fat. They are also involved in the formation of endogenous substances, such as enzymes, hormones, and blood cells [77].

Many B vitamins are mitotropic substances—including vitamin B1 (thiamine), B2 (riboflavin), B3 (niacinamide), and B6 (pyroxine, pyrixodal, and pyridoxamine). The functions of these compounds, which are essential for the body, include, for example, pacemaking in cellular carbohydrate metabolism, inhibition of protein glycosylation, and excitation and transmission of stimuli in the peripheral nervous system by thiamine. Riboflavin is involved in cellular metabolism in the form of the flavin coenzymes flavin adenine dinucleotide (FAD) and flavin adenine mononucleotide (FAM). Here, flavin coenzymes are indispensable for the course of dozens of enzymatically catalyzed oxidations and reduction reactions and, thus, for the intracellular oxidative balance. They are able to act as electron acceptors or donors, depending on the direction of the reaction. Thus, riboflavin performs an essential role in the mitochondrial respiratory chain. Furthermore, it is involved in the metabolism of other B vitamins, including the formation of niacinamide. As a coenzyme component of nicotinamide adenine dinucleotide (NAD) and nicotinamide adenine dinucleotide phosphate (NADP), this vitamin is also involved in numerous enzymatic reactions in metabolism and performs a key role in energy formation in the mitochondria. In this process, it works closely together with $CoQ_{10}$. Among the B6 vitamins, pyridoxal phosphate and pyridoxamine phosphate are particularly important in relation to human metabolism since they assume coenzyme functions in more than 100 enzymatic reactions. Due to the great importance of vitamin B6 in amino acid metabolism, a deficiency of B6 can cause growth disorders and atrophy of the musculature, thymus gland, and gonads, especially in cases of severe deficiency [75].

Vitamin C (ascorbic acid), an organic acid, belongs to the group of hydrophilic vitamins. The importance of vitamin C for our immune system is well known. However, this basic substance fulfills a multitude of other vital functions in the body. Recent research has come to the conclusion that vitamin C is much more important as an antioxidant and especially as an enzyme cofactor than previously thought. For example, as a cofactor, this vitamin plays a key role in the formation of collagen, in immune defense, and in the formation of cerebral hormones and neurotransmitters. In addition to these functions, vitamin C is a very effective radical scavenger in the body [75].

Vitamin E (tocopherol) is a grouping of eight vitamins, all of which are lipophilic in character and have tocopherol activity. Most E vitamins have antioxidant activity [78,79]. This is also one of its most important functions. Through its function as a radical scavenger, tocopherol is able to protect polyunsaturated fatty acids in membrane lipids, lipoproteins and depot fat from destruction by oxidation (lipid peroxidation). In this process, tocopherol itself becomes an inert mesomeric stabilized radical. The tocopherol radical is then reduced to form an ascorbate radical. The ascorbate radical is regenerated with the help of glutathione (GSH) [80].

*Studies on the efficacy of Vitamins*

In addition to the studies on electron/proton carriers, vitamins also show a positive record of use in mitochondrial medicine, particularly in the production of ROS and, thus, protection from oxidative damage.

For example, vitamin E reacts more rapidly with peroxide radicals than do polyunsaturated fatty acid molecules, thus protecting mitochondrial membranes from excessive oxidative damage [81]. In addition, it reduces the production of ROS in mitochondria [82]. Vitamin E also protects against hyperthyroidism. A condition which leads to increased mitochondrial ROS production and, consequently, oxidative damage [81]. It also prevents hyperthyroidism-induced reduction in mitochondrial complexes and ensures that cellular functions are maintained [81]. This could be mediated by the ability of vitamin E to scavenge ROS. Analysis of the effects of electron chain inhibitors on the mitochondrial release rate of $H_2O_2$ suggests that vitamin E may influence the level of autoxidizable carriers. Moreover, vitamin E is preferentially incorporated into mitochondrial membranes [81].

In contrast, vitamin C acts as a mild pro-oxidant that can produce free radicals and consequently stimulates mitochondrial biogenesis [83]. It is also known that vitamin C can be highly concentrated in mitochondria by means of the specific mitochondrial sodium-vitamin C transporter 2 (SVCT2). In this way, it can reduce the proliferation of cancer stem cells (CSCs) by more than 90% in a combination therapy together with doxycycline and azithromycin [83].

Not in humans, but in a spinocerebellar ataxia type 3 (SCA3) Drosophila model, polyglutamine (polyQ)-mediated mitochondrial damage leading to loss of neurons and damage to non-neuronal cells could be successfully treated with vitamin B6 [84]. An abnormality of vitamin B6 metabolic pathways caused by pathological polyQ expression could thus be bypassed. Active vitamin B6 is involved in hundreds of enzymatic reactions and is very important for maintaining mitochondrial activities. In this study, vitamin B6 supplementation suppressed mitochondrial damage in viscera and inhibited cellular polyQ aggregates in fat bodies, indicating a promising therapeutic strategy for the treatment of polyQ [84].

*3.3. Minerals*

Minerals are vital nutrients, usually present as ions or inorganic compounds, which the organism cannot produce itself. They are essential for many functions, such as the formation of bones, the maintenance of osmotic pressure, or the formation of hormones [85]. Minerals, of which 22 are considered physiologically necessary for the human body, are divided into two groups in the body, so-called bulk elements and trace elements. Bulk elements are represented with a higher concentration than 50 mg/kg body weight. Trace elements, on the other hand, have a required concentration of less than 50 mg/kg body weight. As bulk elements are usually ionized in the aqueous milieu, they are referred to as electrolytes. Trace elements, on the other hand, are metals that are absorbed by the body in very small amounts (often only a few micrograms) [85,86].

*Molecular mechanism/Biological function*

An important bulk element in mitochondrial medicine is magnesium ($Mg^{2+}$). It is indispensable for the proper functioning of numerous biochemical processes. The physiological spectrum of action of magnesium is enormous because it is involved as a component or cofactor in several hundred enzymatic reactions. These include, among others, the production of nucleic acids, participation in all reactions triggered by ATP by stabilizing the produced ATP molecule, which is mainly present as a complex with a central magnesium ion [75,87]. Free $Mg^{2+}$ ions influence the potential at cell membranes and act as second messengers in the immune system. They stabilize the resting potential of excitable muscle and nerve cells and the cells of the autonomic nervous system [88]. In addition, along with $CoQ_{10}$, and B vitamins 2 and 3, it performs an important role in the mitochondrial respiratory chain, where it activates diverse enzymes. For example,

the activation of thiamine to the co-enzymatically active thiamine diphosphate requires a magnesium-dependent thiamine kinase [75].

Dietary Mg intake has been shown to be often inadequate in the Western population. This inadequate intake has been associated with a number of adverse health effects, including restlessness, nervousness, irritability, lack of concentration, fatigue, general feeling of weakness, headaches, but also hypertension, cardiovascular disease, muscle cramps, and type II diabetes [87,88]. In addition to bulk elements, such as magnesium, trace elements, such as copper or zinc, also perform an important role in mitochondrial function.

*Studies on the efficacy of Minerals*

The fact that mitochondria have been shown to be able to both accumulate and release $Mg^{2+}$ makes them an important intracellular $Mg^{2+}$ store [89]. Together with recent advances in the field of $Mg^{2+}$ transporter research, which have led to the identification of the plasma membrane $Mg^{2+}$ transporter Solute Carrier Family 41 Member 1 (SLC41A1), the mitochondrial $Mg^{2+}$ efflux system SLC41A3, the mitochondrial $Mg^{2+}$ influx channel Mrs2, and a mitochondrial $Mg^{2+}$ exporter, highlights the importance of gastensium balance in mitochondrial function [89]. $Mg^{2+}$ has been shown to enhance the activity of three major mitochondrial dehydrogenases involved in energy metabolism. While the activities of isocitrate dehydrogenase (IDH) and 2-oxoglutarate dehydrogenase complex (OGDH) are directly stimulated by the $Mg^{2+}$ isocitrate complex or are stimulated by free $Mg^{2+}$, the activity of the pyruvate dehydrogenase complex (PDH) is stimulated indirectly via the stimulatory effect of $Mg^{2+}$ on pyruvate dehydrogenase phosphatase, which dephosphorylates and thus activates the pyruvate decarboxylase of PDH. OGDH functions as an important mitochondrial redox sensor [89].

Dysregulation of these Mg transporters and channels is caused by and also contributes to impaired Mg homeostasis [90]. Thus, decreased levels of free ionized intracellular Mg ([Mg]i) could cause Mg stores, such as mitochondria, to release Mg via SLC41A3 [90]. Decreased mitochondrial Mg levels ([Mg]m) could, in turn, impair further Mg/MgATP-associated mitochondrial signaling and function, which could explain the mitochondrial overproduction of reactive oxygen species (ROS) and decreased ATP levels observed in Mg-deficient mice [90]. Liu et al. recently reported that Mg deficiency in diabetic mice increases mitochondrial oxidative stress and contributes to cardiac diastolic dysfunction [90]. In a low Mg diet-induced mouse model, mitochondrial oxidative stress was also found to contribute to cardiac diastolic dysfunction. Mg supplementation was able to suppress mitochondrial ROS overproduction and reverse diastolic dysfunction, and, therefore, in this case, Mg acts as a mitochondrial antioxidant [90].

The relationship between low Mg status, which is caused by several factors (for example, low Mg intake and absorption, genetic defects in Mg transporters, obesity, type 2 diabetes mellitus (T2DM)) and oxidative stress was also shown by Barbagallo et al. [91].

Thus, low Mg status may trigger increased free radical production (ROS), oxidative damage, and activation of redox signaling. The increased oxidative stress, in turn, can lead to the release of inflammatory mediators, which represent a state of chronic low-level inflammation considered to accompany aging and termed "inflammaging" [91].

### 3.4. Further Mitotropic Substances

In addition to the substances already mentioned, there are now a growing number of other compounds whose efficacy on the mitochondria has been confirmed. These include resveratrol and spermidine.

### 3.4.1. Resveratrol

In recent years, no mitotropic substance has garnered as much published attention as resveratrol, which is present in many foods, e.g., in grapes, red wine, peanuts, and blueberries [92]. Resveratrol is a polyphenolic phytoalexin belonging to the stilbene family. It is a secondary antioxidant plant substance whose phenolic hydroxyl groups have a

high redox potential. In nature, both the more common trans- and also cis-forms exist. In addition, there are also the derived glucosides [92].

*Molecular mechanism/Biological function*

In numerous studies, resveratrol has been found to possess many different bioactivities in the human body, such as antioxidant, anti-inflammatory, cardiovascular, anti-cancer, diabetes mellitus preventing, anti-obesity, neuroprotective, and anti-aging effects [93]. Resveratrol's abilities are thought to be mainly due to its influences on the cell membrane [94]. Similar to $CoQ_{10}$, it seals proton leaks while neutralizing reactive oxygen radicals [95]. In addition, it has the ability to stimulate endogenous antioxidant enzyme systems, such as superoxide dismutase (SOD) [96]. Furthermore, resveratrol induces mRNA expression of age-associated genes [97]. Resveratrol is also thought to lead to a stimulation of peroxisome proliferator-activated receptor gamma coactivator 1-alpha (PGC-1$\alpha$) [98]. This signaling pathway coordinates the expression of additional antioxidant enzymes [99,100].

*Studies on the efficacy of Resveratrol*

The properties of resveratrol in the mitochondrial context make this substance particularly interesting for mitochondrial medicine.

For example, resveratrol has been shown to modulate mitochondrial dynamics both in vitro and in vivo in experimental models, and to increase oxidative metabolic capacity [101,102]. Resveratrol regulates antioxidant enzymes located in mitochondria, reducing the production of ROS by these organelles [102]. In animal models of cardiovascular disease, metabolic syndrome, and muscle disease, resveratrol has thereby been shown to be protective [101]. In addition, resveratrol was reported to protect human adult retinal pigment epithelial (ARPE) cells from chemical-induced oxidative stress and cell death through a mechanism involving mitochondrial protection. Resveratrol also improved the respiratory capacity and rates of oxidative phosphorylation in cells [103]. By upregulating manganese-dependent-SOD (Mn-SOD/SOD2), resveratrol likely protects mitochondrial function through an indirect antioxidant effect. The exact mechanism deserves attention as several transcription factors are involved in the modulation of Mn-SOD expression [103,104]. In addition, the PGC-1$\alpha$-dependent signaling pathway, which is a specific target of resveratrol in mammalian cells, also coordinates the expression of antioxidant enzymes [99,100]. In a model of Alzheimer's disease, Manczak et al. found that pretreatment with resveratrol protected mitochondria of N2a cells (mouse neuroblastoma cell line) from Amyloid-$\beta$ exposure [105]. Resveratrol prevented Amyloid-$\beta$-induced disruption of mitochondrial fusion and fission by maintaining the expression of genes, such as mitofusin 2 (Mfn2), optic atrophy 1 (Opa1), dynamin-related protein 1 (Drp1), and mitochondrial fission protein 1 (Fis1) [105]. For Parkinson's disease, it was shown that these proteins Mfn2, Opa1, Drp1, and Fis1, which are involved in the control of mitochondrial fusion and fission and are downregulated in a rotenone-induced Parkinson's disease model remained at a normal level by resveratrol administration [106]. Resveratrol also had an antioxidant effect by reducing the formation of reactive species. Since impaired mitochondrial dynamics are associated with Parkinson's disease, the use of resveratrol in this context is of great interest [106,107]. Resveratrol has been reported to extend lifespan in several studies. Administration of resveratrol in healthy and mildly obese subjects resulted in higher gene expression and serum concentration of sirtuin-1 [108]. In general, the mechanisms of the prolonging effect of resveratrol are similar to those of caloric restriction [109]. Caloric restriction known to positively affect aging in model organisms [97]. Matt et al. investigated the effect of the caloric restriction mimetic resveratrol on the expression of age-associated genes by analyzing the mRNA expression of SIRT3, FOXO3, and SOD2 in peripheral blood mononuclear cells (PBMCs) treated ex vivo with resveratrol. It was found that treatment of PBMCs with resveratrol resulted in a significant increase in mRNA expression of all genes tested [97].

### 3.4.2. Spermidine

Spermidine is an endogenous, natural substance that was first discovered in male seminal fluid, which gave the substance its name. It is now known that spermidine exists in all body cells [110]. This naturally occurring polyamine, also called monoaminopropylputrescine, is a biogenic polyamine and an intermediate in the formation of spermine from putrescine and decarboxylated S-adenosylmethionine.

*Molecular mechanism/Biological function*

It is involved in a variety of molecular and cellular processes due to its antioxidant and anti-inflammatory properties, including induction of autophagy, apoptosis, DNA stability, and mitochondrial metabolic processes [111,112]. Exogenous spermidine supplementation has been shown to attenuate age-related dysfunction. In this context, spermidine also stimulates mitochondrial biogenesis via the SIRT1/PGC-1$\alpha$ pathway, among others, and exerts its anti-inflammatory effects via mitochondrial ROS-dependent AMPK activation [113–115]. Mechanistically, it thereby shares the molecular pathways used by other mimetics for caloric restriction and induces protein deacetylation [116].

*Studies on the efficacy of Spermidine*

Similar to resveratrol, spermidine also regulates genes responsible for the control of mitochondrial fusion and fission. In addition, Western blot results with B-cell lymphoma 2 (Bcl-2), Bcl-2-associated X protein (Bax), and caspase-3, and NLR family pyrin domain containing 3 (NLRP3), interleukin 18 (IL-18), and IL-1 β, showed that spermidine also prevented apoptosis and inflammation, and increased the expression of neurotrophic factors in neurons of Senescence Accelerated Mouse-Prone 8 (SAMP8) mice [117]. The reduction in anti-inflammatory agents was evidenced in another study, which showed that spermidine reduced the levels of cytokines associated with inflammation (interferon gamma IFN-γ), IL-1β, IL-6, and tumor necrosis factor alpha (TNF-$\alpha$) in mice [118]. The results suggest that the anti-aging effect of spermidine is related to the improvement of autophagy and mitochondrial function [117]. In addition to autophagy, mitophagy, the selective disassembly of mitochondria by autophagy, is also enhanced by spermidine [119]. Spermidine has also been reported to regulate inflammation by inducing anti-inflammatory (M2) macrophages. However, the underlying mechanisms remain unclear. However, it has been shown that spermidine-induced M2 polarization is mediated by mtROS [114]. The concentrations of mtSO and $H_2O_2$ were significantly increased by spermidine, resulting in mtROS-mediated activation of AMP-activated protein kinase (AMPK) and ultimately enhanced mitochondrial function [114]. In addition, hypoxia-inducible factor 1-alpha (Hif-1$\alpha$) was upregulated by AMPK activation and mtROS [114]. This was required for the expression of anti-inflammatory genes and the induction of autophagy [114]. Spermidine-induced M2 polarization was found to be mediated by increased autophagy [114]. In mice in which inflammatory bowel disease (IBD) was induced by dextran sulfate (DSS), the inflammatory status was improved by macrophages treated with spermidine in vitro [114]. Thus, spermidine may trigger an anti-inflammatory program driven by mtROS-dependent AMPK activation, Hif-1$\alpha$ stabilization, and autophagy induction in macrophages [114].

## 4. Conclusions

The preservation of mitochondrial function is a central component of preventive and therapeutic measures within mitochondrial medicine to avoid numerous—especially age-associated—diseases. Research is focusing on mitotropic substances such as $CoQ_{10}$, NAD boosters, but also vitamins and trace elements or substances, such as resveratrol and spermidine, as suitable active ingredients to achieve this. The effect of these active ingredients, which are often specific and improve mitochondrial function, has now been proven by studies both in vitro and in vivo. Nevertheless, there is still a great potential for these agents in terms of their indication areas and the ideal application, which in many parts is not yet fully clarified. The list of enzymatic reactions and signals in which mitotropic substances intervene is long. However, it has been shown that the effect of many mitotropic

substances is based on two important properties. The property to act antioxidant and the improved transport of electrons and protons in the mitochondrial respiratory chain.

$CoQ_{10}$ and $NAD^+$, as electron/proton carriers, naturally possess these properties. Both substances either have a direct antioxidant effect or promote the activity of further enzymes, which contribute to the neutralization of ROS/mtROS. Vitamins can have both antioxidant and prooxidant effects. For example, vitamin E reacts more rapidly with peroxide radicals, while vitamin C can act as a mild pro-oxidant that produces free radicals and consequently stimulates mitochondrial biogenesis. Even magnesium showed antioxidant properties in a low Mg diet-induced mouse model, in that Mg supplementation led to a reduction in mtROS overproduction.

Resveratrol also joins the ranks here. It regulates the mitochondria-based antioxidant enzymes, such as Mn-SOD/SOD2, and the expression of other antioxidant enzymes via the PGC-1$\alpha$-dependent signaling pathway. Additionally, resveratrol, such as spermidine, modulates genes for mitochondrial processes of fusion and fission. Similar to vitamin C, spermidine can also act pro-oxidative by increasing the concentrations of mtSO and H2O2, which resulted in mtROS-mediated activation of AMP-activated protein kinase (AMPK) and ultimately improved mitochondrial function.

The main focus of mitochondrial medicine is the use of mitotropic substances to bypass defective respiratory chain complexes, especially complex I, and modulate the oxidative balance of the cell, thereby activating downstream enzymes and signaling pathways of the antioxidant system. Due to their versatile properties, the mitochondria targeting substances mentioned here represent a significant component for mitochondrial medicine. A component mainly based on antioxidant properties. If these substances and the relatively small arsenal of mitotropic substances were used effectively, the prevention and therapy of acquired mitochondrial dysfunction could already find broad application. Since many drugs affect the mitochondria and mitotropic substances can, in many cases, mitigate the side effects of necessary medication, a mitochondrial-supported adjunctive therapy is recommended as a supplement to classical medicine. In this way, significant improvements can be achieved even in the case of severe mitochondriopathies.

**Author Contributions:** Writing—original draft preparation, D.S.; writing—review and editing, S.W. and J.B.; supervision, J.B. All authors have read and agreed to the published version of the manuscript.

**Funding:** This research received no external funding.

**Institutional Review Board Statement:** Not applicable.

**Informed Consent Statement:** Not applicable.

**Data Availability Statement:** Not applicable.

**Acknowledgments:** The authors thank Franz Enzmann for his scientific advice. This work was supported by the Baden-Württemberg Ministry of Science, Research and Art.

**Conflicts of Interest:** Jörg Bergemann has consulting contracts with MSE Pharmazeutika GmbH, Bad Homburg, Germany.

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
