# Peer review of "Natural Mitochondria Targeting Substances and Their Effect on Cellular Antioxidant System as a Potential Benefit in Mitochondrial Medicine for Prevention and Remediation of Mitochondrial Dysfunctions"

_cimb, doi:10.3390/cimb45050250_

Round 1

Reviewer 1 Report

The review “Mitochondria targeting substances and their effect on cellular antioxidant system as a potential benefit in mitochondrial medicine for prevention and remediation of mitochondrial dysfunctions” tries to give a general overview of substances used for ameliorating mitochondrial disorders.

One of my major concerns is the novelty of the topic, since these substances are known for a long time and are used in the treatment of mitochondrial disorders/dysfunctions with controversial results. Recently several reviews have been published on molecules used in the therapy of mitochondrial dysfunctions that fully discuss the effect of the molecules listed in this review (Viscomi C, Zeviani M. Strategies for fighting mitochondrial diseases. J Intern Med. 2020 Jun;287(6):665-684. doi: 10.1111/joim.13046. Epub 2020 Apr 13. PMID: 32100338. Michio Hirano, Valentina Emmanuele, Catarina M. Quinzii; Emerging therapies for mitochondrial diseases. Essays Biochem 20 July 2018; 62 (3): 467–481. doi: https://doi.org/10.1042/EBC20170114. Russell OM, Gorman GS, Lightowlers RN, Turnbull DM. Mitochondrial Diseases: Hope for the Future. Cell. 2020 Apr 2;181(1):168-188. doi: 10.1016/j.cell.2020.02.051. Epub 2020 Mar 26. PMID: 32220313). In addition, it is unclear whether the authors want to focus the review only on natural substances active for mitochondrial diseases. In fact, they do not name synthetic substances used in therapy or clinical trial such as idebenone or EPI743.

Therefore, if the authors are focused only on natural molecules, it would also be necessary to discuss the possible antioxidant effect of molecules that, for example, stimulate the Nrf2 such as sulforaphane and carnosic acid.

The review has a very general long introduction on the part about the structure and function of mitochondria that is unnecessary, while the description of mitochondrial dysfunctions and mitochondrial diseases is less detailed. Furthermore, the purpose of a review would be to summarize the information in the literature as a table in which are listed the molecules considered, the cellular or animal models in which they have been tested, the concentrations used etc. In this way, the review would become more useful for the reader to obtain the information of interest.

I’m sorry, but in my opinion this review is not complete, is not focused and does not add anything more than other reviews already present in the literature.

Author Response

One of my major concerns is the novelty of the topic, since these substances are known for a long time and are used in the treatment of mitochondrial disorders/dysfunctions with controversial results. Recently several reviews have been published on molecules used in the therapy of mitochondrial dysfunctions that fully discuss the effect of the molecules listed in this review (Viscomi C, Zeviani M. Strategies for fighting mitochondrial diseases. J Intern Med. 2020 Jun;287(6):665-684. doi: 10.1111/joim.13046 . Epub 2020 Apr 13. PMID: 32100338 . Michio Hirano, Valentina Emmanuele, Catarina M. Quinzii; Emerging therapies for mitochondrial diseases. Essays Biochem 20 July 2018; 62 (3): 467–481. doi: https://doi.org/10.1042/EBC20170114 . Russell OM, Gorman GS, Lightowlers RN, Turnbull DM. Mitochondrial Diseases: Hope for the Future. Cell. 2020 Apr 2;181(1):168-188. doi: 10.1016/j.cell.2020.02.051 . Epub 2020 Mar 26. PMID: 32220313 ). In addition, it is unclear whether the authors want to focus the review only on natural substances active for mitochondrial diseases. In fact, they do not name synthetic substances used in therapy or clinical trial such as idebenone or EPI743.

  • This review article is intended to explain the background knowledge of basic mitochondrial mechanisms on the one hand, and on the other hand, only natural substances that can also be listed as dietary supplements are actually addressed. We think that these two points make this manuscript interesting also for people who have only little to do with this topic and at the same time want to have the most important information about the mitochondria and the points of attack addressed therein by just these natural substances. The fact that these are exclusively natural substances has not been elaborated enough so far. This has been changed in the manuscript - See manuscript.

Therefore, if the authors are focused only on natural molecules, it would also be necessary to discuss the possible antioxidant effect of molecules that, for example, stimulate the Nrf2 such as sulforaphane and carnosic acid.

  • To provide only an overview, we have focused on the molecules already mentioned. However, NMN showed positive effects in one study, mediated via the SIRT1/Nrf2/HO-1 signaling pathway. This has been incorporated into the manuscript - See manuscript.

The review has a very general long introduction on the part about the structure and function of mitochondria that is unnecessary, while the description of mitochondrial dysfunctions and mitochondrial diseases is less detailed. Furthermore, the purpose of a review would be to summarize the information in the literature as a table in which are listed the molecules considered, the cellular or animal models in which they have been tested, the concentrations used etc. In this way, the review would become more useful for the reader to obtain the information of interest.

  • As mentioned, this article is intended to introduce non-specialist readers to the subject of mitochondrial energy metabolism and to inform them about the possibility of intervening here with natural substances. The creation of a table would certainly be helpful for one or the other reader but would not lead to any added value in the context mentioned.

Reviewer 2 Report

The manuscript entitled "Mitochondria targeting substances and their effect on the cellular antioxidant system as a potential benefit in mitochondrial medicine for prevention and remediation of mitochondrial dysfunctions" is a pertinent and interesting review in the field of Mitochondrial medicine and mitochondria targeting drugs. This review provides interesting information about mitoceuticals including their Molecular mechanism / Biological function and the studies that support their use. However, some details could be improved in order to be published. 

The introduction section should be adjusted to what is to be discussed in the later sections of the paper. In my opinion, there are entire editorialized parts that could be omitted since this section should lay the groundwork for what is to be discussed later and this is not the case, since it does not provide a clear conclusion that easily lead the reader to the essential point of the review (mitoceuticals).

Quality of figures can be improved, showing more details in line with the main text.

Sections "Mitochondria in the process of aging" and "Mitochondrial dysfunctions – clinical relevance" include relevant information that is described in a very superficial manner. These sections should be further expanded to highlight the relevance of the experimental evidence described in the text. I would suggest complementary tables; particularly, for the section "Mitochondrial dysfunctions – clinical relevance".

Finally, the Abstract section should be changed accordingly. Some sentences in the abstract are quite confusing i.e. "In addition to modern and innovative diagnostics, the most important tool of Mitochondrial Medicine are mitochondria targeting drugs with the help of which an attempt is made to compensate for an existing mitochondrial dysfunction". The conclusion in the abstract should be more clearly and formally stated in accordance with the text.

Author Response

The introduction section should be adjusted to what is to be discussed in the later sections of the paper. In my opinion, there are entire editorialized parts that could be omitted since this section should lay the groundwork for what is to be discussed later and this is not the case, since it does not provide a clear conclusion that easily lead the reader to the essential point of the review (mitoceuticals).

  • The introduction has been changed significantly. This section now leads, as the abstract already does, better to the actual topic - See manuscript

Quality of figures can be improved, showing more details in line with the main text.

  • The figures have been improved - See manuscript

Sections "Mitochondria in the process of aging" and "Mitochondrial dysfunctions – clinical relevance" include relevant information that is described in a very superficial manner. These sections should be further expanded to highlight the relevance of the experimental evidence described in the text. I would suggest complementary tables; particularly, for the section "Mitochondrial dysfunctions – clinical relevance".

  • I agree with you that the topic "Mitochondria in the process of aging" has a great importance in general. However, for this summary, an elaboration of this part would be beyond the scope. In the section "Mitochondrial Dysfunctions - Clinical Relevance" no table was included, because in the studies partly the effect is given with the corresponding disease or else an interaction with a certain molecule. A table would either not make sense or would lead to a rewrite of the entire manuscript.

Finally, the Abstract section should be changed accordingly. Some sentences in the abstract are quite confusing i.e. "In addition to modern and innovative diagnostics, the most important tool of Mitochondrial Medicine are mitochondria targeting drugs with the help of which an attempt is made to compensate for an existing mitochondrial dysfunction". The conclusion in the abstract should be more clearly and formally stated in accordance with the text.

  • The abstract has been revised - See manuscript.

Reviewer 3 Report

This review made a mixed impression on me. On the one hand, it is quite encyclopedic, well written and structured. The exception is the last part - consideration of resveratrol and spermidine is non-obvious. A gigantic number of compounds affect mitochondria, and it is strange why the authors limited themselves to only them. Possibly, it was necessary to separate the compounds according to the mitochondrial targets used for the therapy of diseases, in particular, ion channels (calcium, potassium channels, etc.), MPT-pore, etc. There is no information about mitocans - therapeutic compounds (conjugates of a drug and a mitochondria-targeted cation (like TPP, F16, rhodamine, etc.). Perhaps, together with the first part of the work, this will give a good result and give the necessary novelty.

Author Response

This review made a mixed impression on me. On the one hand, it is quite encyclopedic, well written and structured. The exception is the last part - consideration of resveratrol and spermidine is non-obvious. A gigantic number of compounds affect mitochondria, and it is strange why the authors limited themselves to only them. Possibly, it was necessary to separate the compounds according to the mitochondrial targets used for the therapy of diseases, in particular, ion channels (calcium, potassium channels, etc.), MPT-pore, etc. There is no information about mitocans - therapeutic compounds (conjugates of a drug and a mitochondria-targeted cation (like TPP, F16, rhodamine, etc.). Perhaps, together with the first part of the work, this will give a good result and give the necessary novelty.

  • For better limitation, the focus in this review was placed only on natural substances, which is why resveratrol and spermidine were included here. In addition, only substances were selected which also have a non-directional effect. Anticancer agents such as mitocans, although it is a very interesting topic and would need to be addressed separately, were not included in this process because they are partially selective for malignant tissues. To make this more visible in the manuscript, the section "Mitotropic substances" has been reviesed - See manuscript.

Round 2

Reviewer 1 Report

No further comments

Reviewer 3 Report

The authors clarified the required questions. Work is improved and can be accepted.